# The Enhancement of Creative Collaboration through Human Mediation

**Teresa Varela [1,\*], Odete Palaré [1,\*] and Sofia Menezes [2,\*]** 

[1]  Largo da Academia Nacional de Belas-Artes, Centro de Investigação e de Estudos em Belas-Artes (CIEBA), Faculdade de Belas-Artes, Universidade de Lisboa, 1249-058 Lisboa, Portugal

[2]  Academia Militar, Centro de Investigação, Desenvolvimento e Inovação da Academia Militar (CINAMIL), R. Gomes Freire 203, 1169-203 Lisboa, Portugal

\*  Correspondence: teresa.varela@campus.ul.pt (T.V.); o.palare@belasartes.ulisboa.pt (O.P.); sofia.menezes@academiamilitar.pt (S.M.); Tel.: +351-927-555-131 (T.V.); +351-931-033-294 (O.P.); +351-965-489-837 (S.M.)

**Abstract:** This text presents a reflection on the elements that coinfluence creative processes in learning. This study highlights a specific period in secondary education at the António Arroio Art School in Lisbon, Portugal, developed during the curricular activity Training in Working Context with students of the 12th grade. It aims to identify interpersonal and intercultural relations utilizing active participation and involvement in communities of artistic practice. This research follows an action-research methodology with data collection via observation and interviews with students. The results show that human mediation promotes significant creative collaboration, the construction of one's own identity, and artistic production with others, and it also allows us to perceive creativity as cultural empowerment. Empathy, emotional understanding, and an atmosphere of trust are the factors that students acknowledge as important in the creative process. Freedom and flexibility in creative collaboration practices, promoting autonomous and critical thinking, are also highlighted. Thus, we conclude that values such as mutual respect, solidarity, freedom of expression, and self-help applied in creative practices are crucial in interpersonal communication between teachers and students.

**Keywords:** human mediation; communities of practice; freedom and flexibility; collaborative creativity; creative practices

## 1. Introduction

Currently, according to some authors on the development of the understanding of creativity, the context of art education seems to correspond to the idea that creativity and critical thinking are challenges of the 21st century for educators [1–6], a designation that arises through the need to find new educational strategies involving education and training systems themselves. These changes, associated with the need to find innovative solutions to the emerging problems of an ever-changing society, led Valquaresma and Coimbra to affirm:

> In a society undergoing constant change in which developing and applying new technologies are the driving force of evolution, creativity emerges as a fundamental "tool" for contemporary individuals [2] (p. 132).

If the evolution we have seen in the face of information and communication technologies reveals changes in the ways of seeing the behavior of society, they will also produce changes in the way of life, which in turn will reflect the way we think and create art as well as the individual or collective processes of creation [7].

Education as a protagonist of social transformation allows us to foster students' creative capacity at all educational levels, thus raising creativity to the level of social and cultural value, turning it into a creative challenge for all. Creative proposals are necessary to solve the emergent problems in a society dealing with constant changes and uncertainty [3–5,8–10]. Bahia and Nogueira state that "it is in this sense that the school community (which undoubtedly mirrors the society in which we live) has come to value creativity as a fundamental educational goal" [11] (p. 335). Furthermore, it is in this context that the development of creative practices based on social sharing; the exchange of knowledge; and the acquisition of skills, techniques, and art in Project 1 was presented—"I am who I am"—carried out with 12 students from a 12th grade class in the discipline of scenery design and technologies, part of the specialization in scenery design at the António Arroio Art School in Lisbon, Portugal. In this case study, we tried to investigate some aspects that influence students' creativity. This text focuses on one of them—human mediations and interpersonal relationships—and how they have affected individual and collective creativity. Rodriguèz, quoting Bourriaud, stated that:

> The essence of artistic practice would then reside in the invention of relationships between subjects; each work of art would embody the proposal to inhabit a common environment and the work of each artist, a fabric of relationships with the world that in turn would generate other relationships, and so on to infinity [7] (p. 7).

In this way, teachers promoted a moment of reflection on "who are they?", impelling students toward self-discovery, as underlined by Bruner [12] in "The Process of Education", which proposes learning for self-discovery in experimentation and involvement in activities, generating dynamics of dialogue imbued with the environment where they occur. In turn, these dynamics aim to promote the need to know, understand, interpret, communicate with others, and share unique ways of thinking and acting, contributing to spiral learning. It is in this environment that teachers and students work cooperatively in learning how to think, do, and be, creating spaces that characterize communities of practice, a concept widely developed by the theoretician and pedagogue Étienne Wenger [13,14] in "Communities of practice: learning, meaning, and identity", and are defined as "groups of people who share a concern or passion for something they do and learn to do it better as they interact regularly" [14] (p. 1).

Wenger characterizes communities of practice in three crucial areas: (i) domain, a feature of common interest by which members value their collective competence and learn from each other; (ii) community, where members participate in joint activities and discussions, help each other, and share information; and (iii) practice, through which members develop a shared repertoire of experiences, stories, and ways of addressing problems. Utilizing this appreciation through social sharing of all the participants' interventions, the combination of these three elements constitutes a community of practice. One way of applying this is to relate students' experiences to real practices through peripheral forms of participation in wider communities beyond the school walls, aiming to develop synergies in the midst of multiple learning possibilities [15].

> It is also crucial to acknowledge that activities promoting internal and external experiences provide a greater range of knowledge in lifelong learning processes; life itself is the main learning event.

In Vygotsky's ideas on "development education", approaches to learning prevail, giving greater historical and cultural breadth to human development, in which the transformation of individuals into active and critical participants refers to the achievement of autonomy in sociocultural practices since the moment of identification [16–20]. This freedom of choice, in which students choose to investigate significant themes, leads to a type of education that promotes reflection and dialogue, as Paulo Freire explained, "in a posture of self-reflection and reflection on [their] time and [their] space, [in which] students will be directed towards an integral growth, conscious of themselves, of others, and the world around them" [21] (p. 36).

Recognizing that, over time, educational practices have changed, we observe a concern inherent in the communication established between teachers and students in the sense of identifying students' epistemological positions as an important part of understanding the teaching and learning process [17,18,22,23]. Values such as freedom, otherness, and affectivity should be daily practices to promote an environment of welfare in learning. According to Branco [20], the way teachers expose their practices to students is fundamental for a mutual understanding of the creation of affectivity. Many of the values and beliefs that are carried into the learning environment can establish a positive connection with students or, on the contrary, create obstacles to their active participation. Learning is reduced to a closed-circuit when taking place in an environment where students feel either limited in their activities or afraid to communicate their ideas and are induced to reproduce discourses that correspond to educational systems where the teacher presents content and where students are mere receptors. It is necessary to understand and listen to students about their learning and opinions on how they acquire knowledge and to boost creativity in students [4,5,22,24–26].

This text seeks to demonstrate that, through practices based on social sharing in the exchange with others and the acceptance of divergent thinking, the transactions that take place will contribute to development of reflexive and critical thinking by the student, stimulating autonomy and, consequently, improving performance concerning the quality of socioaffective interactions among all those involved [20,23,24,27,28].

> Here lies the fundamental responsibility of educators in the school environment: the development of students through mediation, in which language is a privileged instrument [27] (p. 20).

Here, too, a need arises to share some reflections on the students' trust, support for decisions, and concerns regarding wellbeing and autonomy, as influential factors and potentiators of collaborative creativity [18,29], which at present is a construct that reflects various trajectories characterized by the multiplicity, transversality, and otherness of the groups and communities where it occurs [1,7–9,11,30], elevating it to an understanding that generates social and cultural empowerment. Furthermore, while it respects differences and individual contribution, collaborative creativity emerges from human interactions within communities and societies themselves, as stated by the authors Gläveanu and Clapp.

> Creativity is nurtured by the diversity of cultural experiences we acquire by participating in our groups and communities. At the same time, it is made possible by culture and enables its growth and transformation, diversifying the range of possible experiences for you and others. This is a vision of distributed creativity, a creativity that brings together people and context, people and material objects, and recognizes their participation and co-evolution in creative activity [30] (p. 56).

This article is organized as follows: Introduction; Materials and Methods, with descriptions and details for later understanding of the results and discussion; Results, divided into two subheadings: the first referring to the importance of personal relationships in communities of practice and the second, implicit in the first, related to the development of creative practices from the dialogical dynamics in the course of the project through the development of ideas and subsequent materialization; Discussion; and Conclusions.

*Presentation of the "I Am Who I Am" Project Theme*

The theme "I am who I am" had two phases: the first corresponding to the activity of Project 1, which took place during the first academic period of 2018/19, and the second together with the curricular activity of Professional Formation, with 120 predefined hours of the annual plan of curricular activities of the 12th year for a total of 30 lessons at the António Arroio Art School, which took place during the second period. The aim was to observe how the students would embrace the project and to understand if the theme was appropriate and interconnected with their interests [17,18,31,32], providing participation and involvement on the part of the students, elevating them to a commitment of self-regulation in the

making of the various activities, activities oriented to elaborating three-dimensional objects under the pedagogical team's responsibility of teachers: Carla Isidro, Rita Anahory, and Teresa Varela.

Initially, this project focused on knowledge consolidation and skills learning. The first activity suggested learning to handle and apply subjects and materials that the students had not yet worked with, in particular, in the area of scenography, where knowledge about the use of wood and its derivatives and the application of metals was transmitted, and in the area of costumes, where the students were able to acquire skills to make fabrics and other textile materials. In addition to improving their technical and artistic skills, teachers suggested that students develop a piece that would convey who they were.

At a later stage, the focus was on the development of skills at the level of autonomy and self-regulation of students' actions, particularly in the relationship with other people and with entities outside the school [23] and in the commitment and responsibility to the search for solutions and concrete realization of real situations, inserted in a work context.

Throughout this partnership, the students made pieces with artistic character, reflecting the contacts established with the partners, namely the resident artists of Pavilion 31 of Júlio de Matos Hospital in Lisbon, and with the Portuguese artist Pedro Cabrita Reis. At the end of the school year, the pieces elaborated by the students were integrated into a joint group exhibition with the resident artists of Pavilion 31. Curiously, the students proposed to continue the theme of Project 1 in the 2nd period—"I am who I am"—in the development of activities with the partner entities in training and work contexts. There was a clear desire for the students to continue to transpose into artistic pieces thoughts about "I am who I am" and to realize these combinations of ideas with those emerging from recent contact with partners outside the school community. In the dialogue with the teachers, there was a need to be free in their choice of meanings that they wanted to instill in their plastic manifestations. There is an appropriation of reflections on themselves that have persisted in revealing who they were, with complex interconnection results as the understanding of creativity and its creative processes [1,4,7–9,30]. In this second moment, which constitutes the main focus of this reflection, students developed their activities in an environment of wellbeing with freedom and flexibility, favoring progressive learning in terms of their personal and social, transversal, and specific competences together with the development of their creative, technical, and artistic skills. The three-dimensional pieces produced in this educational environment result from a plastic exploration that questions new ways of thinking to seek solutions to problems interconnected with others, becoming an encounter of interactions, meanings, and transformations [7]. The human mediations that took place throughout the activities provided experiences and practices that enabled an understanding between the actors of the communities of practice, based on support for decision-making of the students, respect for divergent opinions and choices, and constructive dialogue for joint learning, resulting in the enhancement of collaborative creativity [18,23,29,33]. In this context, the main research issues arose:

- What contributions and interlinkages stand out among the participants as influential elements in collaborative creativity throughout the process?
- What are the mediations that are evident among students in the elaboration of artistic production throughout the process?
- To what extent do values such as freedom and flexibility in classroom management influence creative collaboration practices?

## 2. Materials and Methods

Intending to demonstrate some of the coinfluences that exist in the processes of individual and collective creativity through human actions, Project 1—I am who I am, was presented as an experimental theme. This research focuses on the contents of personal interest with educational subjects, the interweaving of transversal competencies from factual information that the student possesses, including perceptions and the transformation into new interpretations. The latter, of plastic

manifestation with meanings that matter to the students, entails communicating their ideas with others, making them concrete, making decisions, and engaging in their own learning [3,17,18,24,34] in the search for understanding and systematization of the interconnections that coexist in creative practices and that constitute agents that enhance collaborative creativity. In this sense and since the study considers the researcher's characteristics and the relationship with the actors, the methodology applied was action-research, with a qualitative-interpretative approach [35,36]. This research aims to understand students' progress; development of their creativity; actions; reflections on divergent opinions; and relationship with others, teachers, and colleagues [22,23,25,26,28]. Based on previous research on the preferences of how students claim to acquire learning better, they stated that activities based on an active construction of understanding are more interesting than activities that recognize students as passive receptors of information [24]. This preference of students for a constructivist learning environment is in line with significant learning, as there is recognition of the importance of the student's "voice". This study gives relevance to students' ability to comment on their learning. However, it is necessary to consider the dialogical dynamics that influence the relationships between teacher and students, as each one develops its own perspective of the teaching–learning process [3,16–18,22,23]. It is a priority to establish efficient dialogue between teacher and student. The transactions arising from this communication are reflected in the relational processes between the process actors, namely, with the interactions carried out with the partners in Training in Work Context, a fundamental connection between the individual and creative collaboration [23,37].

After authorization from the school administration, the teachers' informed consent was shared, constituting the teaching team, and informed consent was given to the pupils involved. The information was communicated to all participants, namely the study's intention, the right to withdraw from the study at any time, the guarantee of anonymity, and the protection of the data collected. This project was carried out with 12 students; however, with the authorized participation of 9 students, high involvement and participation were observed.

The data collection was carried out using classroom observation, documentary collection (portfolios relating to the two moments of the project), audiovisual support (photographic recording throughout the activities), 12 daily questionnaires prepared over 3 weeks, and semi-structured interviews with teachers (4) and students (18), which mostly covered the period of 120 hours (equivalent to 10 weeks) corresponding to the curricular unit—Training in Working Context. The interviews were conducted outside school hours thanks to the availability and collaboration of all those involved. After the data collection, a qualitative and interpretative analysis of teachers' and students' observations and perceptions was carried out, with the present text showing the students' indications and comments regarding their experiences and perceptions in the learning processes [38–40].

## 2.1. Data Collection Tool and Procedure

For this study, the methodology of action-research was applied in a dynamic that allows for understanding and analysis of a set of interactions that occurred during the learning processes, making use of the information collected through observation, informal dialogue, and interviews conducted with some students throughout the various activities for a qualitative analysis [38,39].

The activities were accompanied, with observation records, informal dialogues and, also, questions were asked to the participants of the activities, in two different moments: 1st interview, at the beginning of the actions in Training in Working Context (which corresponded to the conclusion of the Project 1- I am who I am, during the 1st school period) and 2nd interview, after the realization of the three- dimensional pieces and their public presentation in the collective exhibition with the resident artists of Pavilion 31 of the Júlio de Matos Hospital, in Lisbon (held at the end of the school year).

Participants answered a set of closed and open questions (semi-structured interviews) to understand the factors that students considered most relevant for their development, as well as their creative processes. In the first interview, within the framework of the theme "I am who I am"—Project I, the questions focus on collecting data and obtaining information about perceptions of

creativity, which are influential elements in their creative processes and understanding of teachers' strategies when presenting and developing activities (class management). In the 2nd interview—a set of different questions, possible indicators are presented, influencing the creative processes and the way the activities took place (with or without flexibility, with or without freedom).

Some factors, also mentioned in studies by Alencar [4] and Oliveira [28], were recognised by students as being the most relevant to their personal development and the drivers of their creative practices. This data survey, relevant in terms of getting to know the perspectives of the students regarding their learning processes, was also referenced in other studies by Kinchin [24], Krapp [32] and Slot [31]. The questions took into consideration theoretical and empirical studies [4,23,29] on facilitators and inhibitors of creativity in the educational context.

Some of the questions, closed and open, are presented in this study:

(1) In the learning processes, do you feel that the activities, guided and accompanied by the teachers, followed a flexible or rigid pedagogical methodology? Yes or no? Explain.
(2) Are the teachers' guidelines motivating and encouraging for students to experiment and learn?
(3) Do you think that teachers' guidance, throughout the learning process, has contributed to the evolution of your creative skills? Yes? No? In what ways?
(4) Do you consider constant contact with your colleagues to be an asset?
(5) During the development of your work, can you mention any aspect in which you have felt collaboration, mutual help from your colleagues?
(6) What skills have you acquired during the course of this work? Social, personal, transversal and specific expertise?
(7) What changes can you observe in your colleagues after this work has been carried out?
(8) For you, what differences were there between the exercise Project 1—"I am who I am" and the work carried out in the context of Training in Working Context (FCT)? Identify the most obvious ones.

## 2.2. Data Analysis

After collecting the data, a qualitative and interpretative analysis of the observations and perceptions of teachers and pupils was carried out. The questions were analysed using the analysis of content, following the guidelines of Amado (2014); there was the interpretation of contents from interview excerpts and the attribution of titles to achieve a classification of the units of meaning. These units of meaning brought together various categories and subcategories, with the present text showing the indications and comments of students regarding their experiences, perceptions and actions in the processes of learning with meaning [38–40].

## 3. Results

As mentioned above, the results are based on qualitative and interpretative analyses using data collection tools centered on surveys (questionnaires and interviews), student portfolios, and the observation and monitoring of activities. It is intended to understand and interpret how students approached the activities, absorbed information, and reflected on perceptions in the acquisition of knowledge throughout creative practices. In this collected and systematized information, the results analysis approximated several categories and subcategories to be investigated, for which the following stand out:

## 3.1. Importance of Interpersonal and Personal Relationships in Communities of Practice

The pedagogical team launched the challenge "I am who I am" to the students together with some references: a documentary and an interview with the choreographer Marlene Monteiro Freitas, which inspired reflection around this theme, and the reading of several texts, namely "I am" by Marta Medeiros, "Thinking" by Vergílio Ferreira, "It is my fault" by Florbela Espanca, and excerpts from "The Life of Forms" by Henri Focillon. Based on these references, the students began their research,

revealing a high interest in exploring themes related to self-knowledge. As a result, the group of students quickly became involved in a more intimate and private reflection on "who were they?"

### 3.1.1. Social Relations in the Class (Teacher–Student and Pupil–Student)

In this environment of reflection and of sharing information and knowledge, students tried to get answers and to share ideas and actions based on motivation, curiosity, and interests, which allows establishing communication between all the actors. It is stimulating for mentors to seek out and research the most significant themes, and there is also a need to build skills in the creation of dynamic processes which favor cultural empowerment, accompanying and mediating the various phases of activities [5,16,19,26]. The diagram presented in Figure 1 illustrates the interpersonal and intercultural relations between communities of practice [13,14] and involves external partnerships (inserted in an educational context), becoming collaborative creativity agents [23,29].

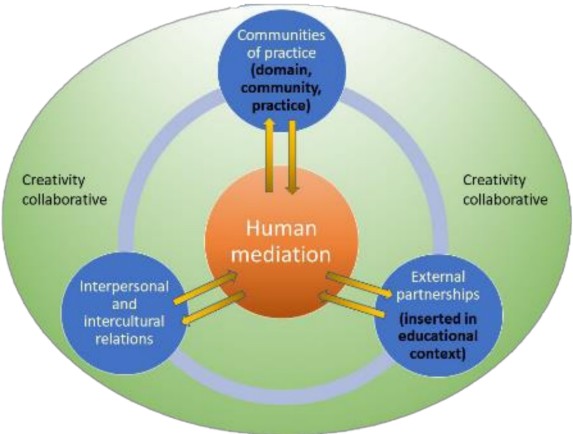

**Figure 1.** Dynamics enhancing collaborative creativity.

In this dynamic relationship, in the constructive dialogues during the classes and through the interviews with some students on personal and interpersonal relationships within the communities of practice, particularly in the relationship between students and teachers, some factors have been identified in this process as contributing elements to the development of their creative processes, namely empathy, emotional understanding, and atmosphere of trust.

Student AL003, in her first interview, states: "I felt that, whenever we talked about the problems of our work, the teachers knew exactly what we were talking about. That was good" (AL003, 2019, 00:25:27, interview 2). This student indicated satisfaction when she realized that the teaching team followed the evolution of her work, allowing establishment of a social and emotional relationship between pupil and teacher which supported the student's development, demonstrating the ability to understand the students' needs, feelings, and concerns about something that occurs (emotional and empathy competence). On some occasions, it has been possible to identify the students' emotional state, thus contributing to improved wellbeing through changes in their activities' performance. For example, the account below illustrates the emotional understanding of students' states of mind at certain times:

> There is also this emotional concern. [The teacher said:] "Look, if you are not well today, do not go to the workshop and stay in the part of the project. [It is] really important so that you can also do a good job, right? There is no [...] authority [saying you have to] do it! Do it!"
>
> (AM004, 2019, 00:26:45, interview 2)

In the social and emotional relationship between pupil and teacher, this sensitivity makes it possible to identify the emotions and to recognize the relationship between them and individual experiences to give them meaning. Another AD008 student, in the second interview, refers to the following:

A personal difference that I found was also during Work Context Training. I already knew the teachers. I already knew the people. This allowed me to have more confidence regarding what I wanted to do, more security, and not to doubt so much my choices.

(AD008, 2019, 00:31:36, interview 2)

This comment recognizes factors that contribute to an atmosphere of trust, referred to by students as elements that help their creative processes and personal development. During the course of the activities, within the sensitivity of each student, the concern arises to establish visual communication through drawing, with graphic characteristics on varied surfaces presenting signs, writings, sketches of their main ideas, and schemes, some with textures and plots, in meeting a structure that seeks to communicate. Proposals emerged. The teachers asked questions to help them reflect, accompanying them and respecting the individual objectives through achievement, experimentation, and error. Thus, this journey's rhythm was established, filled with the differences introduced by each one, in regular meetings, respecting the individual and the collective, suggesting possible solutions to the issues listed, subject to presentations and debates in the community contributing to reaching conclusions together. In these communities of practice and in interpersonal and personal relationships, it is observed that collaboration, mutual help, and the sharing of experiences between the students themselves influence the development of their activities, giving rise to discoveries, as one student says in her first interview:

Colleague X also had a very complex job [...], and sometimes, [when] I was waiting for something in my work, I would help her, so it was also good to learn to work with the materials she used.

(AL003, 2019, 00:31:58, interview 1)

Interactions between colleagues also provide learning about other subjects and materials. Participation in something that is not part of your own work enriches the student on a personal, social, and specialist level. Also, student AC004 said in her second interview: "you do a little of my [work], I do a little of yours" (AC004, 2019, 00:18:03, interview 2). In other words, the students discovered that the exchange of experiences in artistic development and the execution of work throughout the activities are an integral part of their creative process.

In the exchange of ideas and through the observation of each other's work, students understand other circumstances, understand other choices, and discover new sensations and different perspectives of observing, accepting each other and their equally valid opinions, characteristics that contribute to the construction of each individual's identity. In these questions interacting with colleagues, we find a social and emotional relationship and a relationship between peers. Interaction, learning to know others, and understanding them better in the space of social sharing in an environment of wellbeing generates more information about oneself and others. The way they work with each other and how they act and react in conversations allows students to reflect on their work, to discuss it, and to present it in greater involvement and active participation in the individual learning process. Another student mentioned the following in her second interview:

I asked X what he thought about how I was doing the arm: "Is it all right? Is it bad?" X told me what he thought [...], and I think it makes us rich. X did the same thing, and it makes us rich.

(AC005, 2019, 00:20:00, interview 2)

The dialogue between colleagues and the solicitation of colleagues' opinions without fear of observation and judgment in mutual help encourages academic development because the basic skills for academic performance, learning, self-esteem, and social skills development are improved. It has also been found that the help offered between colleagues improves skills, security, and autonomy in self-regulation processes. In the same sense, problem-based learning as a learning strategy in peer classes allows for increased technical vocabulary and critical judgment and improved problem-solving

skills, value, and group work prioritization. This concern in supporting and monitoring each other's work shows a collective growth, as student AM004 says:

> All the help [...], all the concern we all had [...], we were in the middle of the workshop and...: "Look, have you been able to do that yet?" [...] "Have you been able to recover that?" [...] Since we were all doing this process of [...] accompanying each other, so it is also funny to see our object grow and objects of others.
>
> (AM004, 2019, 00:42:54, interview 2)

These statements underline the importance of fostering dynamics that generate socially shared knowledge, which influences creative practices. For this reason, the active participation of all those involved in communication and interaction with others and the environment without denying the value of the individual and his or her individual mind enables the creation of artistic manifestations resulting from conceptual, technical, and plastic conjugations, which reinforce discoveries at the emotional, intellectual, and creative levels. In this dynamic, the teacher occupies a central role since they can encourage, promote, and create the appropriate space for collaboration and knowledge construction.

### 3.1.2. Elaboration of Artistic Production and Self-Knowledge

In an environment of shared interests, with committed participation of the group in making themselves known, one begins to observe the involvement, seriousness, and honesty with which the students present their thoughts, ask questions, talk about their research, and declare their intentions. Thus, during the first moment of approaching the theme self-discovery and experimentation, the students began to present their reflections and to later elaborate volumetric pieces that translated their ideas.

In one of the works, a student presented a graphic composition where she registered different phrases with bright colors, from which we highlight the following "I am what I am even when I do not know what that is" (Portfolio "I am who I am," 2018). In this work, the sentences occupy the bottom of the page, and in the center, an image was glued—a female figure taken from a magazine—in which the hair is represented by wool threads of various colors that fill and animate the figure. Through the search to "know whom she is," the student represented from the free graphic expression what she considered to be "a part of what I am, but also of what I want people to associate me with," written on another page of the student's portfolio. The bottom of the page reveals the student's thought—the perception of the difficulty of defining who she is. At the same time, she used vibrant and cheerful colors to convey her taste for life. The female figure's costume, whose dress has some phrases, reads "Mamma sei la più Bella nel Mondo," highlighting family's importance. This work reveals the first relationship between themselves and their own family, considering the students' ages, between 16 and 18. Besides, adolescents seek to carry out and to engage in activities of their own interest, both inside and outside school, a process that contributes to the formation of their identity [19,24,31], determining factors in the students' own intentions of creative expression [23,28,41,42].

In another work, the student, among the references given in class and her research to develop it, drew abstract figures like faces searching for a way to transmit "I am who I am." She began by drawing lines, establishing paths and crossings, and creating movements, with the expression in the line she printed revealing several faces, intending to communicate that the human being is constantly changing. In her personal life, this student is characterized by her energy, which she channeled into various activities, namely dance. The student intended to create a symbol of a constantly changing bust, and so she built a bust consisting of a central conical volume and several loose elements (eyes, nose, ears, and mouth). The central volume, which is about 70 cm high, contains multiple perforations that allow the aforementioned elements' position to be changed. This project's initial proposal foresaw the public's interaction with the piece, changing the loose elements' position, intending to create several visual conjugations, and resulting in multiple faces. In this work, the student revealed her intention to build for others to intervene, which means that it is a work for others in the context of human relations

and about who she is. This can send out a signal that each of us is built up by relationships with others and that others also affect us, creating connections and changes [33]. These transactions were expanded in the 2nd moment approaching the theme, with the beginning of curricular activity during the Training in Working Context, which took place at the end of the 1st period. In this context, to relate the project to the work context, a visit was made to the Júlio de Matos Hospital (H.J.M.), Pavilion 31 of the Art Gallery to meet with the Artistic Director Sandro Resende, our partner. This meeting allowed us to meet and observe some of the works of resident artists and to talk with some of them about holding a group exhibition at the end of the school year. It was also possible for the students to meet with the artist Pedro Cabrita Reis, who received us informally in his studio and at home, to discuss the paths of arts and artistic practices. These contacts made a crossing of different realities possible, enriching all those who participated, resulting in pieces that plastically translated the interconnections experienced.

For example, in another work, the student tried to transmit her thoughts about herself and her individuality based on what is primordial to her, something she says she took from her conversation with the artist Pedro Cabrita Reis, who stressed the importance of what is primordial to her artistic work. Furthermore, in this work, the student mentions the following:

> I took this question of the primordial and began to develop it with several references [...] to the Triadic Ballet and the Butoh art and another reference to constructivism. All these references are part of a construction, and the idea of construction is that it is primordial in my work. This is always present. I am supposed to play with this question of the primordial that is not seen, and the references I sought were purposely so that what I consider my individuality would not be visible. They are beliefs related to the limitation and the way the form manages to be perpetuated and limited in a certain artistic work. [...] I couldn't remain indifferent to all the small steps that I took with everything provided, so I tried to play with it.
>
> (AM02, 2019, 00:01:09, interview 2)

In her work, entitled Manifestation of Being, the student embarks on this game of experimentation and construction, using various materials—clay, iron, styrofoam, and pulp, among others—elaborating a form that somehow transmits "I am who I am", not in words but through the plastic construction of the object. In this sense, from the above-described references and with the materials mentioned, it creates a human figure in profile based on one of the Triadic Ballet costumes and consists of a base and an iron axis where various elements stand out. A bicycle wheel is in the center, from which several sticks and a foot made of polystyrene and pulp are broken. Above this wheel, there is an aluminum sheet with an almost elliptical shape fixed with rivets and a clay head. In the lower part of the piece, under the bicycle wheel, a composition is made up of metal strips fixed to the piece's base, inspired by Vladimir Tatlin's Monument to the Third International detailed with different metals and cotton thread. The most curious thing about this piece is that the student claims not to give a visual form that defines it and, as such, tried to respond to the work by constructing her piece plastically.

> Organicity and poetics are present in the idea of construction, which I consider myself to be. In this project, I end up falling into the contradiction of trying to define who I am. In sculpture, its elements define me, neither visually nor by their symbolism: they only relate to the need to limit myself to a form.
>
> (portfolio, AM002, 2019)

From the references that interested her visually combined with the diversity of experiences lived, the absorption of this information, and the reflection on the participation in the activities carried out in the communities of practice, her creative process emerged. In this way, developing the students' different work processes will constitute creative practices as constructing their own research on creation processes [41,42]. All the synergistic forces have contributed to transdisciplinary, expansive, and distributed knowledge by creating artistic productions that reflect the communities

where they are inserted and where creativity enhances social and cultural empowerment [23,30]. The students developed individual work; however, the discussion of ideas; the explanation of intentions; the construction in collective with others; and the establishment of an open dialogue between students, teachers, and external partners in a working context generated a free, informal environment that evolved together.

*3.2. Educational Strategies and Classroom Management*

This subject's curricular activities, usually based on a rotation system, were divided into three components—project, scenography, and costumes—developed through an organic dialectic between conception/development of ideas and practice/materialization.

In learning, the focus was on developing students and their creative practices with flexibility and freedom through working with each one [5,18,22,24,28]. In the encouragement of research and self-discovery, the conquest of new perceptions and understandings about different ways of "making it happen" progressively developed the cognitive and metacognitive capacities of the students alongside their affective and motivational capacities (autonomy, confidence, motivation, responsibility, and self-regulation) [3,23,29,37]. Together with a greater breadth at the conceptual, technical, and artistic levels, these characteristics can contribute to increased security and confidence in students' decision-making and the knowledge and experience acquired in the area of specialization [3,18].

Through this mediation centered on collaborative learning, at a given moment, the need began to arise for some students to move, in time, from the conceptual field to the material field with different rhythms. The student began to organize themself, to make choices, and to ask "how to resolve this form?" or "how to resolve this moment? It is recognized that this transition to the actual construction of the idealized form without concrete perception of results is not easy. The fact is that these negotiations between conceiving, recording on paper the idea, and moving on to the actual practice of constructing a form are not watertight, so it is necessary to give students time to think, create, test, construct, adapt, and change.

Freedom and Flexibility in Creative, Collaborative Practices

Students must understand that there's no problem going back on their choices. If a material does not fit the object, it is always possible to find other resources that will allow them to communicate the idea and the thought they want to convey. Sometimes, hazards become solutions, where some unforeseen events can provide results that are eventually assumed in pieces. For example, in another work, the mistake was assumed and became the solution—in a piece built with a volumetry similar to a mushroom base, about 90 cm high, the student idealized a painting in shades of black and yellow, represented in a succession of rings; first, she painted several rings in shades of black, and then, she applied color in shades of yellow in the intervals where there was no black shade, but when she painted the shade of yellow, the student did not remove the excess paint on the brush; she quickly observed that the paint dripped in certain places of the piece, but when she observed the visual effect, she chose not to correct it, as it conferred a greater organic character to the object, something that she intended to transmit.

Faced with dilemmas so naturally inherent to learning processes, the student must be allowed to experiment; explore; make mistakes; and, if necessary, make mistakes again until he or she can make better mistakes. In these diverse and distinct experiences, the teacher's guiding role is nothing more than a learning facilitator, an agent who enables the student to express themself and to develop their knowledge, who questions them to help in their evolution, and who shares knowledge and tools that enable them to empower themself as an individual. In this way, activities promote the freedom to think, choose, and decide to register greater participation and greater involvement, according to the collection of information from informal conversations and interviews with students. Student AL003 refers to the importance of freedom of choice in creative processes, allowing her to express meanings that she is interested in conveying when she refers to the following:

Freedom in terms of, for example, telling us that *you really have these solutions, you can choose which one you want....* This is also a very important thing considering that [...] it is a creative process [...] and, this is so inherent to each one.

(AL003, 2019, 00:28:00, interview 1)

It can be said that creative freedom embodied in freedom of expression happens when pupils support their decision-making.

The theme "I am who I am" was quite liberating since its purpose was to give students the freedom to create, following their thinking, commitment, and responsibility in their actions and combining the necessary processes to achieve these same thoughts. Students have known and identified potential combinations inherent to these same experiences; they have understood through constructive processes that the characteristics and properties of each matter or material employed influence the achievement of the final form of their artistic products. Thus, with these students' intention to express plastically what they wished to communicate, experiments and explorations of materials were carried out and materials were combined, searching for multiple ways of solving a situation. This flexibility of choice and diverse works is seen when student AL003, in her first interview, stated that:

I was thinking of making all the flowers the same. [...] Then, I was offered ways of making the flowers that were not as I had imagined and not making all the shapes the same, and the work became much more interesting.

(AL003, 2019, 00:30:39, interview 1)

In this situation, the student recognized that the possibility of exploring other solutions has enabled her to gain knowledge about new ways of realizing her ideas by expanding her knowledge. As her colleague, student AC005, says:

We want to do one thing and come up with ways and different ways of doing that thing. Maybe I already think differently from last year [...]; maybe I can already cover other different ways of doing and using new materials.

(AC005, 2019, 00:24:15, interview 2)

All these experiences allow the addition of knowledge and give more artistic and personal confidence to the students when choosing the most appropriate solution, according to their thoughts' concretion, contributing significantly to their creative processes. In a coexistence between mental and material processes, rhythmic as tides, in a flow and reflux of human events, the creative processes have made it possible to open up horizons, to test options, and to check aesthetic and operational choices in order to understand how ideas work and have been understood by students in an enriching way through human mediations. In this context, learning becomes effective in this exchange of ideas in the observation and dialogue between all that diversified solutions arise in the face of a situation.

Students also recognize the importance of constant contact with colleagues and teachers in a collective sharing, both inside the classroom and outside, to develop their interpersonal and intercultural relationships, as described by student AM004 in her first interview:

My colleagues [...] were central to how we all develop a different way of thinking, the crossing of these thoughts, and also does critical thinking.

(AM004, 2019, 00:31:17, interview 1)

At a certain point, students become aware of the importance of their actions and know that their opinions and ways of acting affect others since there is room to communicate their ideas and to listen to others' ideas, in mutual respect, even at the disagreement of the other. These aspects inherent in human mediations provide new perspectives and new ways of accepting that change and constant adaptation to emerging situations are necessary for growth and evolution. Participation and collaboration among all will be necessary to generate reflections and discussions on how learning is achieved. Student AM004 confirms this in her first interview:



> Facing not only [...] the image I have of the works of others but I also know what is behind them and all the explanation of [something] that exists and [...] all the help that was also given to me in the constructive process and all this sharing of knowledge, not only in the face of the works but also in the face of the people we are and [...] in the sharing of space with which we also come into contact, with each person's behavior towards the things that happen. Therefore, it is a learning experience in terms of opinions and access to new theories and everything else but as a behavioral thing of knowing how to act, knowing how to be.
>
> (AM004, 2019, 00:32:34, interview 1)

and identifies its learning as

> An adaptation in all aspects [...] all the situations that arise for us [...], not only the behavioral ones, the interactions that we have with people but also the constructive ones, but in the end, it was that adaptation in all aspects and knowing how to extend me a little to it. In other words [...], this thing of stagnating, there is some comfort in it, knowing that no matter how this situation arises, we are, we continue to be, but we will extend ourselves a little more. [...] I think that is what I learned.
>
> (AM004, 2019, 00:37:45, interview 2)

These statements above reveal that the challenges presented to students throughout the activities developed in these communities of practice inherent to the circumstances gave rise to transformations in their perceptions and interpretations of what surrounded them, convincing the students to participate actively, benefiting the whole group both at the level of learning competencies and at the level of developing autonomy, critical, and creative awareness with the added value of observing growth in collective. Student AM004 feels that:

> We are all [...] with different sobriety about things. [...] I think we have all grown a lot. [...] the way we even talk about things is really [...] different and [...] doing all this time perspective: well, we were at this point, and now we are doing this. [...] It is rewarding [...] because I feel that not only have I grown, but we have all grown together.
>
> (AM004, 2019, 00:41:51, interview 2)

In this sense, the communications were established with the partners in Training in Working Context. The realization of visits to exhibitions, the conversations with some resident artists of Pavilion 3 of H.J.M., as well as the informal conversation with the artist Pedro Cabrita Reis were fundamental contributions in the implementation of this project. They were determining factors in the students' work in creating multiple and differentiated objects that reflect the resources and means involved and in establishing communication bridges between the various synergies provided throughout the activities. The interdependent activities of the collaborative actions are reflected in the artistic practices through transdisciplinary processes which build meaning in their plastic manifestations and the environment where they arise. Some of these pieces are significant for the students, and others have performative characteristics and/or are installations; however, they all reflect the context of real construction, integrated into a collective exhibition held at the end of the school year.

## 4. Discussion

### 4.1. Importance of Interpersonal and Personal Relationships in Communities of Practice

When we refer to the importance of interpersonal and personal relationships in the communities of practice, previously mentioned in dialogue with the results presented here, two types of coexisting relationships are identified: the interconnections expressed by the students in the teacher–student and pupil–student relationships, and the moments highlighted in the elaboration of the artistic production and self-knowledge of the students.

### 4.1.1. Social Relations in the Class (Teacher–Student and Pupil–Student)

During the classes, the students' recognitions of influential contributions and interconnections in the creative processes were noted. Based on the data collection, they identified the interrelationships between pupil and teacher and between pupil and student as important. In these communities of practice, both teachers and students shared opinions and demonstrated divergent views; this enabled an exchange of ideas and a flow of dialogical dynamics that led to a growing collective sharing, which brought about a common wellbeing.

The importance of fostering learning experiences that stimulate the participation and motivation of students with meaningful activities helps to promote emotional and intellectual wellbeing [3,18,22,23,29]; indeed, the interest and unexpected curiosity of students in finding meaningful ways and means that somehow convey what they wanted to communicate were remarkable. The commitment, participation, and responsibility in the execution of the activities, in the informal dialogues that took place, and in the interviews showed that an environment that promotes trust, respect, and empathy contributes to self-esteem and autonomy in the students. This was underlined by O'Toole: "when individuals can choose what and how they learn, they are asserting their need for autonomy and creativity" [18] (p. 76).

In their relationships with teachers, students have recognized three key factors in developing their creative processes: empathy, emotional understanding, and an atmosphere of trust. These factors enabled mentors to demonstrate their sensitivity by putting themselves in others' shoes and, in so doing, they were able to respond correctly to students' emotional reactions. Empathy as emotional competence is achieved when active listening is combined with emotional understanding and assertiveness at a behavioral level. It also involves understanding the changes from one emotion to another and reflecting on the emergence of sometimes contradictory feelings. In an environment of trust, where wellbeing is essential, it is necessary to consider that students' emotional involvement is distinct from cognitive involvement, and the latter often depends on the former [3,20,28]. Moreover, both influence the development of creative capacity, the effective-emotional dimension, and the dimension of students' cognitive and metacognitive processes [18,22,23,37]. Also, Chua et al., mentions "that trust based on affection, instead of trust based on cognition, is fundamental to connection between individual differences in cultural metacognition and creative collaboration" [37] (p. 117) because trust based on emotions and affections encourages the sharing of new ideas that drive creative collaboration.

However, without forgetting the implications inherent in the incentive and presentations by which students learn how to learn and the need for continuous teacher training [3,10], we cannot neglect the importance of values and beliefs in the influence of educational strategies that teachers employ in the classroom [3,10,20,24,26,28]. As Branco [20] also explained, they determine the dynamics between teachers/learners and students during interpersonal communication activities. Here, the teacher's fundamental role emerges as a driving force in developing quality, social, and affective relations with students and, consequently, stimulating and contributing to creativity in practices. Alencar stresses that many teachers are unaware that the extent to which creativity flourishes depends largely on the environment. They also do not know that the ability to create can be expanded by strengthening attitudes, behaviors, values, beliefs, and other personal attributes that predispose the individual to thinking in an independent, flexible, and imaginative way [4] (p. 48).

Branco reinforces this: "teachers must understand the importance of their daily interactions with students and one another in order to promote relationships of trust, cooperation, autonomy, creativity, and self-development" [20] (p. 47). It is precisely in this space of communities of practice that reflection and intervention with others occur, discovering in social relations that they establish mediation in the process of cultural appropriation, guaranteed by the involvement in learning, linked to wellbeing [3,18,23,29,33]. The correlation between these characteristics allows good functioning in progressive learning, valuing the interactions and collaborative experiences. In realizing the work in spaces shared by all, the collaboration of mutual help is registered. With the observation of what others are developing and sharing experiences that also influence the understandings, perceptions,

and new understandings that emerge, discoveries are made. In addition to greater connectivity with others, learning about other subjects and materials takes place. For example, by suggesting that the student experience what another colleague is producing, exchanging functions momentarily, students experience different actions which, although not related to their specific project, allow them to broaden their knowledge. Furthermore, consequently, they develop their plastic and artistic capacities, inherent to their creative processes, in dialogue with each other. The sharing of a common space where the collective feels free to interact with the other in the exchange of ideas in acceptance but also in confrontation with divergent ideas promotes a dialogue with respect for the other and enhances the social and emotional relationship between peers (the social relationship between student and student). According to the constructivist theory of learning, this is a construction of the individual due to interaction with the environment. However, the mere relationship with the environment will not be enough to achieve knowledge; it will be necessary to generate confrontation and divergence to result in more effective learning. Therefore, in these dynamics, students find space to think and reflect on their actions, freely and flexibly, encouraging them to share socially. Agirre considers that the student should become aware of others: artist, spectator, and critic, among others. With their colleagues' and teachers' knowledge, the student can learn from other works to return to their work and to enrich it [8]. As for reflection and self-reflection on others' opinions, it helps create autonomous and critical thinking. The importance of contribution of others in the construction of individual and collective cultural identity is emphasized by Bandura when he states that "The achievements of a group are the product not only of the sharing of knowledge and skills of its different members but also of the interactive, coordinating and synergistic dynamics of its transactions [33] (p. 75). Students need to understand that considering other opinions, learning to make choices, and actively participating in their learning processes allows them to become more autonomous, creative, and critically aware of their actions, which materialize in an improvement of their life experiences and shared understandings in the collective sphere [20,23,29,30].

### 4.1.2. Elaboration of Artistic Production and Self-Knowledge

Concerning the perceptions resulting from the creative processes verified in conception and plastic realization, there were three distinct moments: first, a relationship of the student with themself; second, a relationship in the construction with others; and third, in the understanding of the artistic production resulting from the interconnection of the interests and experiences lived by its participants, that is, the understanding of creativity as sociocultural empowerment.

In the first approach to the theme "I am who I am," the students revealed a relationship with themselves, a more personal reflection on themselves, with questions and doubts, as student AD008 says, "The concept itself, I spent much time meditating on what the concept of the play "I am who I am" would be, what I would like to work on" (AD008, 2019, 00:00:00, interview 1).

In this reflection, transposition to the characteristics of their works their personality and interests and correlation with important meanings for them were explored, where "We were more focused on thinking and trying to discover who we are. […] We had to restrict ourselves more to ourselves and research within ourselves and what makes sense to us", as described by student AC005 in her second interview.

Considering the ages of the students, between 16 and 18 years old, teenagers try to carry out and get involved with their interests inside and outside the school. Family and friends also play an important role in forming their identity, determining factors in the students creative expression intentions [4,10,16,28,31]. These are just some influential factors, so we cannot overlook others' importance, the internal and external motivations, and the surrounding context, namely in interaction with others. Moreover, in this relationship with others, the second moment verified in work developed by some students is addressed. Recalling the work done by a student in building a bust with loose elements (eyes, nose, good, and ears), this student made a self-reflection on what it is like to be connected with others, revealing the other's importance to help define who they are. For this reason, her piece is

an invitation for otherd to intervene and to modify the bust according to the loose elements' position. This recognition of the individual through experiences in socialization and interactions with others and in the exchange of knowledge will lead to actions; reflections; and the elaboration of new learning, new knowledge, and creative manifestation [41]. The position is also defended by Stetsenko when quoting Vygotsky, referring to social interconnection as an essential characteristic for the progressive evolution of the human being, linked to the development of one's own creativity, since he stated that "the whole future of humanity will be achieved through creative imagination" [43] (p. 45).

Artistic production is no longer an object of contemplation only, where the spectator simply visualizes the artistic work; it happens and requires our participation, interaction, and construction with others [7,23,29,30,33]. Therefore, the relationship with the spectator has also been altered, leading to an interconnection between them and the work made by the artist, as Rodriguèz defends.

> A work done by the artist [...] is in the interconnection, game, and participation with the spectator. The artistic creation takes place, not in the artist's mind acting under an inspiring muse, nor in the eyes of the public, impacted by a particularly special work, but in the encounter between the two, in the relationship, in the duration of the game that gives existence to work [7] (pp. 5–6).

In other words, creativity is understood as an interactive, multifaceted, and distributed phenomenon that connects us with other people and the context in which it happens [1,4,7–9,11,30,41].

Furthermore, here, we refer to the third moment of mediation that contributes to collaborative creativity when the students' work results from all the synergies imbued in practice communities. Transactions are carried out regularly in a common interest with an appreciation for otherness and collective learning in a social sharing where all are agents for joint growth [23,29,33].

In the student's work, in the presentation of her piece entitled Manifestation not particular to being, she manages to produce and limit her reflection on "I am who I am" in a certain plastic work. However, she says she does not identify with it. She simply found a way to respond to the request, playing with "the duration of the game that gives existence to work", limited to a form. She said she was not indifferent to all the experiences she had experienced and to the contacts established throughout the activities and the inferences that resulted, which resulted in the process of experimentation and construction transposed in her piece. All these factors, combined with her own personality, and social and cultural identity proved to be essential in her evolution as an individual, in the development of her creative potential, and in the achievement of artistic production. This intrinsic relationship between the creative processes and the contexts in which they occur is currently reflected in the artistic understanding and societies as Rodriguèz defends when he states that artistic practices have abandoned the conventional character attributed to the artist in order to attribute to creators the capacity to "generate contexts and transdisciplinary processes of construction of meaning and production of meaning and experience, leaving behind their merely aesthetic role to become agents of a function of social catalyst" [7] (p. 6).

In this sense, the actors contribute ideas, ask questions, give divergent opinions, participate and get actively involved, and place creativity as agents that increase social and cultural empowerment. Gläveanu and Clapp underline that "creativity as cultural empowerment is a vision that respects individual contributions, placing them within a broader framework of community and society" [30] (p. 60).

Education, as a mirror of communities and societies has a preponderant role because, through its structure and forms of human organization, and their interconnections, it allows us to promote creative capacity, thus raising creativity to the level of social and cultural value, making it a creative challenge for all [1–10], a challenge already affirmed by several authors and also by Oliveira when he states that it is necessary that "teachers be made aware of the relevance of stimulating creativity in students [because] there are many challenges and problems of the contemporary world that lack creative answers." [28] (p. 90).

## 4.2. Educational Strategies and Classroom Management

Freedom and Flexibility in Creative, Collaborative Practices

Educational practices that encourage interaction, participation, and dialogue with others and the world around them can enable individuals to act consciously and critically to transform differences into resources to develop new sociocultural forms and practices. Gadotti stresses that

> To educate means, then, to enable, to empower, so that the learner can seek the answer to what he or she asks, means to form for autonomy [...]. His method: dialogue. It is the disciple who must discover the truth. Therefore, education is self-education [44] (p. 10).

From the beginning of the activities, the pedagogical team wanted the students to work freely, to choose what and how they would like to work, and to make individual decisions, such as what themes and materials to choose and what meanings could be understood behind the plastic expression from their objects, in this orientation, of the activities that take place from interests that the students have, wish to discover, and seek to investigate and that the teachers sought to understand, respect, and support [3,10,18,24]. As some authors point out, the importance of supporting students' decision-making, emotional understanding, constructive dialogue, and positive reinforcement in the development of students' activities is determining factors for the learning of students' social, personal, and transversal competences and of competences interconnected with the development of self-esteem, autonomy, confidence, creativity, and reflexive and critical thinking [4,8,16,24,29,45]. Also Niza stated, "We believe that dialogue is the method applicable to help reflection, to enrich proposals, to raise solutions as to how much there is to rethink about Education" [45] (p. 42). It is in this spirit that the dialogic dynamics in creative practices, based on values of mutual respect, solidarity, free expression, and mutual help, enable learning for life, reinforcing the collective agency described by Bandura [33]. In these mechanisms of human action, through the interactions between those involved, the process develops from which the whole community of practice will unfold, reinforcing collaborative creativity through interpersonal and personal relationships. These networking relationships enable interconnection of multiple areas of knowledge, promoting a transdisciplinary learning according to Santos:

> Transdisciplinarity maximizes learning by working with images and concepts that mobilize together the mental, emotional, and bodily dimensions, weaving both horizontal and vertical relationships of knowledge. It creates situations of greater involvement of students in the construction of meaning for themselves [46] (p. 76).

The students' motivation, attitude, and commitment in the face of experiences and worries throughout the learning process were visible, with the concern of getting the artistic objects to communicate and transmit what they wanted, something common to all but belonging to each one. A reformulation through open and collaborative experiences stimulates the appearance of others and new interpretations and meanings, contributing to the development of the capacity for reflection on the self, others, and the world around them. In this sense, allied activities in a work context also make it possible to know other contexts [15,23] in an evolutionary progression of the human being, conscious of the intention to "consolidate and expand the partnerships that link us to the educational, cultural, artistic, and business world, which allow on the one hand to promote a systematic adjustment of the training that we give to our students, but also to expand our social contribution" [6] (p. 6).

Thus, in the course of creative practices, in the meetings and conversations held with partners in training in a working context, namely with the Artistic Director Sandro Resende, together with visits to individual exhibitions of some of his artists, the students learned that the circumstances as well as the commitment in the realization of their objects of artistic production were conditioned to a demand that went beyond individuality, since it involved a collective responsibility. The continuous contact with the artists of Pavilion 31 and their unique and meaningful ways of expressing made the students

aware, challenging them to develop their potential; to deepen their reasoning, in a constant concern to communicate with others; to transmit their ideas; and to reach forms and volumes that would plastically express their creative thoughts. At this point, technical and artistic knowledge reinforces individual and collective responsibility, corresponding to the dynamics mentioned above in socially concerned communities of practice.

By reflecting on their actions and how they involve others, the students acted responsibly in the face of situations, decided on procedures, carried out technical-artistic processes appropriate to their projects, and made adaptations when necessary, contributing to their self-knowledge in valuing work processes, involving students in their learning processes developed their autonomy, motivation, and self-esteem. Therefore, the importance of educational strategies that integrate intertwining of areas of knowledge through partnerships outside the school is recognized; "in this way, these competencies and qualifications often become more accessible if those who study have the possibility to test themselves and enrich themselves by taking part in professional and social activities in parallel with their studies" [15] (p. 20). Therefore, it is necessary to create synergies between inside and outside of the school community to construct knowledge. As Sullivan defines it, knowledge is "an exclusively human process that results from interactions and dialogue leading to new understandings" [41] (p. 1188). Also, Santos states that "knowledge is conceived as a network of connections", characteristically transdisciplinary, from the multiple articulations of areas of knowledge [46] (p. 75). In this sense, learning is the process by which knowledge is created through the transformation of experience and artistic education practices are encouraged in conjunction with activities in a real context in order to reflect and establish a relationship of greater communication between the school community and the local, family and beyond. These are some of the factors influencing individuals' creative potential [23,28].

The intention of an artistic manifestation capable of achieving something significant for the students generates a production commitment for some aspect that goes beyond itself, especially for others and for society [19,23,30]. In these reflections on the experiences lived, the emotions reflected before other realities, other contexts that, at a given moment, interconnect and cross the paths of the students' learning, lead them to adapt themselves through the circumstances and to transcend themselves beyond what is known. Transformations and adaptations in the face of established coexistence and contacts; new commitments; and responsibilities related to their actions, ways of thinking, and acting are inherent in the communities of practice where they are inserted [29,41]. Paulo Freire also mentioned ideas when he addressed the question of relationships with the world in the affirmation that there is a plurality in the very way of proceeding or thinking. This existence is nourished by the capacity to "transcend, discern, dialogue (communicate and participate)," where "to exist is individual, yet it is only accomplished when in relation to others, in communication with them" [21] (p. 40). The same author explains that humans are beings of integration and that, through the related relationships between experienced situations, there is integration and not stagnation.

Here too, the central importance of the teacher is defended, for if every human being is naturally creative, the environment in which learning takes place is an influential factor in the flowering of creativity in which the teacher must act as a facilitator of practices that develop the capacity to create and strengthen independent, flexible, and imaginative ways of thinking on the part of students, enhancing creative expression [4,5,10,25,26,28]. As Alencar states,

It is known that the phenomenon [creativity] is complex, multifaceted, and multidetermined. Its expression results from a complex network of interactions between individual factors and variables in the socio-historical-cultural context that interferes with creative production, with an impact on creative expressions, on the opportunities offered for the development of creative talent, and also on the modalities of creative expression, recognized and valued [4] (p. 48).

The teacher assumes a relevant role in that he or she is a promoter of activities that encourages participation and offers affective quality in interactions and social relationships within educational contexts, integrating practices oriented to the enhancement of collaborative creativity, where each one

matters to the whole collective [3,16,20,23,29]. As Collard and Looney point out, teachers make significant changes in their approaches to learning and the way they saw their own roles. These approaches to learning include working beyond the classroom and school, using student experiences and work as a teaching and learning resource, open expression of emotions, valuing collective work, open learning opportunities (where the answer is not yet known), using the body and all senses, and engaging with the community at large [23] (p. 353).

> People reflect the communities and societies that they integrate, so, as Bandura states, "people are in part the products of their environments, but in selecting, creating, and transforming their environmental circumstances, they are also producers of environments" [33] (pp. 75–76).

## 5. Conclusions

In these communities of practice, the empowerment of creative collaboration takes place through human mediation as well as by stimulating divergence in the way students learn to think about themselves and the world around them and by trying to understand who they are, in contact with other people, encouraging students to participate and work together in finding solutions to individual and collective goals. In this study, it has been possible to obtain some information that has helped us adjust the research issues mentioned above.

Educational strategies based on learning that value freedom, flexibility, constructive dialogue, and support in students' decision-making reveal themselves to be generators of students' own dynamics. In these dynamics, they question and learn collectively, and participate actively and are interested in their own learning process, referring to a reflection and self-reflection on others' opinions, promoting autonomous and critical thinking.

This study indicates that students' results depend on their affective-motivational relationship with their teachers, whether it is in following up activities or in their sensitivity and ability to understand emotional states, creating empathy and an atmosphere of trust, which translates into stimulating more active participation and greater emotional involvement in their creative practices.

In the course of the creative process, the incentive to present and debate ideas about the evolution of the work; the suggestion to observe, question, and follow each other's work; joint living in workshops, where the objects idealized by each student are produced but in collective growth; the sharing and exchange of experiences; discovery in the experimentation of materials and their multiple combinations; and curiosity about the work of the colleague, constitute interconnections that highlight the importance of the student–student relationship. Moreover, in this space of the communities of practice through the interventions of each one, in reflection, acceptance, and confrontation with divergent ideas, individual and collective knowledge are expanded. This social and emotional relationship between peers enables a greater understanding of themselves and each other, how they work, how they interact with each other, mutual help, and collective concern that promotes greater confidence in the students, self-esteem, and self-regulation of the work to be done.

This study also shows that the pedagogical approach to developing creative practices based on mutual respect and individual freedom, so that the student manifests and expresses themselves, and chooses and decides what they want to explore, allows us, as teachers, to accompany and support this enthusiasm in meaningful learning that makes sense to the students. Topics of interest with greater proximity to the students' lives developed in an environment of motivation and stimulation where the contents are apprehended through this social participation in the mediation of tools, rules, commitments, commitment, and performance in the actions, with freedom in decision making in the processes to be carried out, for the achievement of the students' objectives come to substantiate the statements of the authors mentioned and of the interviews carried out to the students regarding the development and greater affirmation of their thoughts, critical awareness, autonomy, and creativity. The students liked to be free, to decide what to do or not, to explore and choose, to create their own organism in the self-regulation of their work, and to promote self-development.

Values such as freedom, otherness, and responsibility prevent the student from realizing the ideas presented within the possible resources and promote active participation and greater involvement and commitment. In addition to artistically manifesting itself, this project has made it possible to establish an organic dynamic with enriched practices by engaging those involved in the activities, which has favored and enhanced collaborative creativity.

The sharing of knowledge and cooperation, the understanding of different thoughts, the acceptance of critical opinion, and the ability to reflect and learn from colleagues and teachers improve perceptions and practices, revealing creation as a collective and intersubjective process for cultural engagement and a balanced social life.

Characteristics that contribute to a broader understanding of creativity, creative practices, and creative processes are still a theme that needs more research and focus on other factors that influence the development of creative potential. Only a few have been addressed in this text. The need to investigate and promote creativity in the human being is understandable, and in this sense, it would be interesting to carry out more research that values the "voice of the student" in order to be able to identify other influential elements in creative practices in order to undertake possible transformations in artistic education practices that can contribute to development of the creative potential of individuals, intrinsic to the global development of the human being, in the construction of its sociocultural identity.

**Author Contributions:** For the present text individual's contributions from the autors were: Conceptualization; methodology; software; validation; formal analysis; investigation; resources; data curation; writing—original draft preparation, T.V.; writing—review and editing, S.M.; visualization, O.P.; supervision, O.P. All authors have read and agreed to the published version of the manuscript.

**Funding:** This research received no external funding.

**Conflicts of Interest:** The authors declare no conflict of interest.

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
