# Peer review of "The Enhancement of Creative Collaboration through Human Mediation"

_education, doi:10.3390/educsci10120347_

Round 1

Reviewer 1 Report

In the reviewed article, the authors have provided a very thorough description of creative collaboration through human mediation using the action-research methodology. It is a very interesting and up-to-date research area, and - in my opinion - this kind of analysis is needed. However, this article is messy, illegible, and needs to be refined, and its scientific side should be strengthened. Throughout the article, it is not clear which elements relate to the knowledge obtained from the literature and which from the action-research. Besides, in the introduction as well as in the rest of the text, there are some recommendations, which should be the subject of conclusions. It also requires greater embeddedness in the theory of mediation and the theory of collaboration.

Moreover, the introduction is inconsistent and has too much content that the authors also write about in later chapters. I would suggest moving the paragraph that starts with "For this study, the methodology of action research ..." (p. 2) to the methodological chapter. Content that begins with "During this dynamic, students find space ..." until the end of the introduction (p. 2-3) I would suggest moving to the end, to the part discussing the obtained results.

The introduction should indicate the need for research. This is not in this article. What exactly is the purpose of this article? Is the searching for answers to the 3 research questions (p. 6) the goal? If yes, why these questions are in the chapter with results, but not in the Introduction section? What previous research has been done in this area? Do they show any research gaps? The reference to the existing literature in the undertaken research area is insufficient.

In the Methodology section, the purpose of the PhD and references to the PhD are not needed. They bring nothing to mainstream research. In the 4th line of the Methodology, there is "demonstrate/exhibit?". I would suggest the authors make up their minds.

In order to organize the Methodical part, I would suggest that the authors divide the content into a) research characteristics, b) research process (step by step), research context (the final part of the Introduction, from "During this dynamic ..." to the end). Currently, it is difficult to assess whether the research was carried out with methodological rigor.

In chapters 3 and 4, the results of the research are intertwined with references to the literature. These parts would be more transparent if chapter 3 contains only the results of the research together with their interpretation by the authors, and chapter 4 compared these results with the literature. This can be difficult because the theoretical background that points out the research gaps in the studied area is insufficient. This would reveal what new scientific knowledge this research brings. Furthermore, what exactly are the practical implications of this research?

In the Conclusion, I would also recommend a clear explanation of the added value of the research and analyses carried out.

Author Response

Good afternoon,
Thank you for yout time and comments on reviewing the manuscript, The enhancement of creative collaboration through human mediation.

In the expectation of answering properly the questions mentioned, I present the uploaded document Review 1_7 november (docx).

Thank very much for the comments!
Best regards from the team,

Reviewer 2 Report

First of all an interesting paper, solid research and might be a good paper after a major restructuring.

First part of the paper is not very clear, it takes time to understand what this project is about and the research questions are presented after many pages. I also ask the author to consider fewer research questions and to reformulate them for the article to be more focused. Too many findings, brief discussions and theoretical framwork are not presented very well. I suggest a much better structure (IMRAD) and to be more focused on one or two research questions instead of three. Long sentences with many detailes can aso be improved by reformualtions and shorter sentences. The research is solid, the author is eager to discuss and conclude, but with a better structure and more focused, this paper will be much better. Remember also to contextualize with the name of the country which I guess is Portugal. Almost there, only pay attention on better structure and focus! Leave some for yor next article.

Author Response

First of all, my deep gratitude for the attention given to the revision of the manuscript.
You point out specific issues that are so importante for us.

I’m sorry, but I have no program to modify, include notes or respond next to your comments. Therefore, we kindly ask you to consider our answers about the following document (docx.)

We have carried out a major restructuring of the article, in the expectation of properly clarifying the issues mentioned in order to turn the initial manuscript into a good article.

Thank you for your precious comments about manuscript!
Best regards from the team,

Reviewer 3 Report

Review “ The Enhancement of Creative Collaboration Through Human Mediation”

The design of the project proposed is a bit simple, but is very interesting.

Some of the arguments lack references. I would suggest the authors adding more citations to support their statements.

Some possibly helpful references on:

Perello-Marín, M. R., Ribes-Giner, G., & Pantoja Díaz, O. (2018). Enhancing education for sustainable development in environmental university programmes: a co-creation approach. Sustainability10(1), 158.

Yeh, Y. C., Chang, H. L., & Chen, S. Y. (2019). Mindful learning: A mediator of mastery experience during digital creativity game-based learning among elementary school students. Computers & Education132, 63-75.

Ciumas, C., Urean, C. A., Muresan, G. M., & Armean, G. (2017). Education And Employment Rate In Romania. Annals of Faculty of Economics1(1), 81-86.

Chua, R. Y., Morris, M. W., & Mor, S. (2012). Collaborating across cultures: Cultural metacognition and affect-based trust in creative collaboration. Organizational behavior and human decision processes118(2), 116-131.

Fulop, M. T., Tiron-Tudor, A., & Cordos, G. S. (2019). Audit education role in decreasing the expectation gap. Journal of Education for Business, 94(5), 306-313.

The literature references should be edited.

Author Response

Good afternoon,

We thank you for your time, comments and helpful references and, in the expectation that we respond with improvements necessary and mentioned, we present the following information in the word document uploaded. 

Thank you very much!

Best regards from the team, 

Round 2

Reviewer 1 Report

Thank you for the opportunity to read again the results of analyzes on the impact of interpersonal relations on creative collaboration. This article has been improved substantially and now it meets the requirements for scientific publications. The discussion part indicates the contribution of the considerations to the collaboration theory and to the practice of action of educational units. In my opinion, it can be accepted for publication.

Reviewer 2 Report

Thank you so much for detailed description of how you understood and interpretated my comments and recommendations in my first review. You have really done a great job and with your new structure, shorter scentences and more focused,  this article is much more readable and very interesting. Well done. Some sentences can still be reformulated and made shorter, but all in all the article is good!